# Transform[ing] heart failure professionals with Indigenous land-based cultural safety in Ontario, Canada

**Javiera-Violeta Durán Kairies**[1,2☯], **Emma J. Rice**[1,2☯], **Sterling Stutz**[1,2☯], **Sharon W. Y. Tan**[1,2☯], **Anne Simard**[3,4☯], **Heather Ross**[3,4☯], **Angela Mashford-Pringle**[1,2☯] *

**1** Waakebiness Institute for Indigenous Health, Toronto, Ontario, Canada, **2** Dalla Lana School of Public Health, University of Toronto, Toronto, Ontario, Canada, **3** Ted Rogers Centre for Heart Research, University Health Network, Toronto, Ontario, Canada, **4** Peter Munk Cardiac Centre, University Health Network, Toronto, Ontario, Canada

☯ These authors contributed equally to this work.
* angela.mashford.pringle@utoronto.ca

**Data Availability Statement:** Data cannot be shared publicly because there may be some identifiable phrases in the talking circle data. Data are available from the University of Toronto

## Abstract

Cardiovascular disease is a leading cause of death worldwide, with disproportionate impacts on Indigenous Peoples in Canada. In Spring 2022, a land-based learning program was piloted and evaluated as an Indigenous cultural safety training for professionals at a cardiac care centre and university in a large urban city. Baseline and endline surveys showed an increase in knowledge of Indigenous histories, cultures, and practices; increased reflection on positionality and intention to create change; and strengthened relationships with the land. Future work should explore the long-term effects of land-based cultural safety training on participant behaviours, and health outcomes for Indigenous Peoples.

## Introduction

Cardiovascular disease (CVD) is recognized by the World Health Organization (WHO) as the globally leading cause of death [1], and by Statistics Canada as the second leading cause of death in Canada for the past 22 years of monitoring [2]. First Nations, Inuit, and Métis people (Indigenous Peoples) in Canada experience disproportionately higher rates of CVD than the general population [3–6] and show increasing incidence, even while the general population incidence is declining [7]. These CVD rates must be understood within the context of the heightened health risks faced by Indigenous Peoples resulting from colonization and colonial policies, assimilation, forced relocation, and historical trauma, as well as present-day racism, discrimination, and social and economic inequalities [7–9].

CVD has physical, mental, and emotional effects requiring extensive and complex care. From 2007 to 2017, CVD, stroke, and vascular cognitive impairment accounted for a combined 2.6 million hospitalizations, with 40% readmitted for related events [10]. The effective management of CVD requires medical, pharmaceutical, mental, and lifestyle interventions [11] with the support of a range of health care providers. A number of national reports and inquiries have documented the extensive racism, discrimination, violence, and deeply rooted

Research Ethics Committee (contact Office of Research Ethics, University of Toronto, McMurrich Building, 2nd floor, 12 Queen's Park Crescent West, Toronto, ON M5S 1S8 416-946-3273, ethics.review@utoronto.ca) for researchers who meet the criteria for access to confidential data.

**Funding:** The On-the-Land Indigenous Education Program was funded by 2021 TRANSFORM HF Collaboration Starter Grant awarded to the P.I., Dr. Mashford-Pringle. TRANSFORM HF is funded by the Ted Rogers Centre for Heart Research and the University of Toronto [see https://transformhf.ca/]. The funders had no role in the study design, data collection and analysis, decision to publish, or preparation of the manuscript.

**Competing interests:** The authors have declared that no competing interests exist.

colonialism that Indigenous Peoples face when navigating the health care system [9, 12, 13]. For Indigenous Peoples with CVD, a rapid review found delays in receiving care and worse long-term health outcomes to be the most prominent disparities, when compared to non-Indigenous people in Canada [14]. However, delays in seeking and reaching care, and worse short-term outcomes were also present and disproportionately affecting Indigenous Peoples [14].

In 1996, the Royal Commission on Aboriginal Peoples put forward Recommendation 4.7.8 that "Staff of non-Aboriginal service agencies directly involved in Aboriginal service delivery be given cross-cultural training delivered by Aboriginal people and organizations and that government funding agreements reflect this obligation" [12, p. 237]. The Truth and Reconciliation Commission echoed similar messages in Calls to Action #23 and 24, which demand cultural competency training for all health care professionals, from students to practicing professionals, across sectors, and with specific inclusion of "Aboriginal health issues, including the history and legacy of residential schools, the United Nations Declaration on the Rights of Indigenous Peoples, Treaties and Aboriginal rights, and Indigenous teachings and practices. This will require skills-based training in intercultural competency, conflict resolution, human rights, and anti-racism" [15, p. 3]. While doctors and nurses are named explicitly by the TRC, all staff involved in cardiovascular disease care for Indigenous patients, including health care provision, research, and administration, contribute to culturally safe encounters with the health care system and thus would benefit from cultural safety training.

By definition, cultural safety training must go beyond skills, attitudes, and knowledge about Indigenous Peoples (cultural competency) and sensitivity to differences (cultural sensitivity), to teach about the complex social and historical contexts and power imbalances which shape interactions, and facilitate self-reflection by learners on their own positionality and its effect on their work [16]. Cultural safety has gained visibility since its introduction by Māori nurses in New Zealand in the 1980s as a response to address disparities in Māori health [17], and cultural safety training has since taken on many forms: online modules [18], lectures, groupwork, workshops, volunteering, land-based learning, and many more [19].

Land is integral to Indigenous pedagogies as such, "land-based learning is a powerful decolonizing tool that centres and honours Indigenous relationships with the land and all of creation" [20, p. 3]. For many Indigenous Peoples, ways of knowing and being are inextricably connected to traditional territories through languages and cultures [20–22] and thus learning cultural safety for Indigenous Peoples also connects back to land. Land-based learning as a method of cultural safety training centers Indigenous worldviews and methodologies, using relationships between humans and the natural world as pedagogy [21, 23, 24]. As land-based learning becomes more widely used as a cultural safety training method, there is a need to understand its effects and effectiveness on the knowledge, attitudes, and behaviours of participants, especially with regards to individual and systemic, present and historical power and privilege. While there are a number of evaluations on cultural safety interventions across health fields, there are few land-based cultural safety training interventions, especially in Canada, directed at health professionals, and even fewer evaluations thereof [19, 25]. In 2022, the University of Manitoba piloted a 13-week cultural competency training for the university library staff which included a one-day session at Turtle Lodge in Sagkeeng First Nation [26]. The relevant question on their evaluation received feedback that participants overwhelmingly enjoyed the land-based learning day [26]. Similarly, a cultural immersion component of a cultural competency curriculum for physicians at the John A. Burns School of Medicine at the University of Hawai'i received extremely positive feedback and high ratings for their speakers [27]. However, there is a need for evaluations on the content, skill-building, and effectiveness of land-based learning for cultural safety training, as well as evaluations on the long-term

impacts on Indigenous patient experiences and outcomes that have been called for in the larger cultural safety training literature [28–30].

## Materials and methods

The PI, Angela Mashford-Pringle, is an Algonquin (Timiskaming First Nation) Assistant Professor and Associate Director of the Waakebiness-Bryce Institute for Indigenous Health, Dalla Lana School of Public Health, University of Toronto. She was born, raised and lives in the Tkaronto area (Treaty 13).

Ethics approval was obtained from the University of Toronto Research Ethics Committee (Protocol #_41340).

From May 31, 2022 to June 2, 2022, the On-The-Land program was held at Hart House Farm. There were 17 participants from TRANSFORM HF, with various professional and educational backgrounds including engineering, medicine and public health. Participants included clinicians (physicians and nurses), university faculty, researchers, program staff and trainees (Master, PhD and Post-Doctoral) who were recruited via emails sent through the TRANSFORM HF network using their internal mailing list, newsletter, and was promoted on social media.

The program comprised of the following topics: introducing yourself, smudging ceremony, overview of Indigenous health, case studies on urban, rural and in-community First Nations, Inuit and Métis health, racism and 3Ps (power, privilege and positionality), and three sessions with Elders that included lived experience with heart failure, and traditional healing with a medicine walk.

Prior to the start of the program, participants were sent a baseline survey via email to complete before arrival. A talking circle with participants happened on the final afternoon to answer 3 questions on what the participants learned during the program. Finally, an endline survey was sent after the end of the program in mid-June. The baseline and endline surveys were hosted on REDCap. Both surveys had a consent form embedded and no identifiable information was collected due to the small sample size and maintaining the confidentiality of each participant. The surveys included questions on knowledge, relationships, self-reflections with the land, Indigenous Peoples, land-based learning, Indigenous health, and their own relationships to these themes. The survey questions were formatted for open text, multiple choice, and Likert scale answers. There was no financial compensation for participants. The baseline survey was completed by 16 participants and the endline survey was completed by 12 participants.

## Results

A total of 16 participants completed the baseline survey and 12 completed the endline survey. With 17 course attendees, this represents a 94% response rate for the baseline and 71% response rate for the endline. Most participants attended all three days of the program.

### Knowledge of Indigenous histories, cultures, and practices

There is a shift in knowledge and awareness that participants had between the baseline and endline surveys, which indicates that they are on their learning journey. Self-rated knowledge of Indigenous history in Canada, colonization and its impacts on Indigenous issues, Indigenous cultural protocols, traditional medicines, terminology regarding Indigenous Peoples, OCAP® principles, and Indigenous data sovereignty increased after completing the land-based learning course (Figs 1–4).

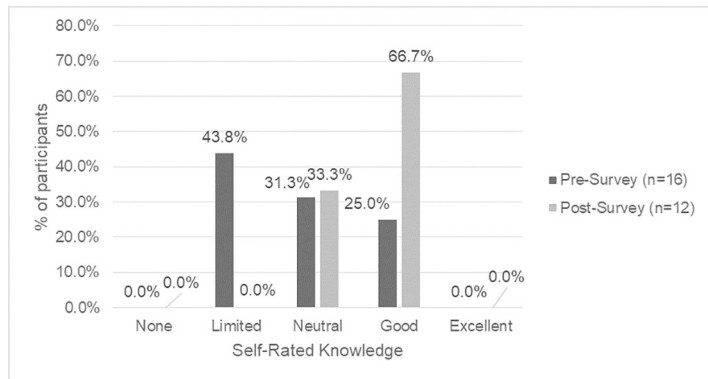

**Fig 1. Baseline vs. endline survey self-reported knowledge of Indigenous history in Canada.**

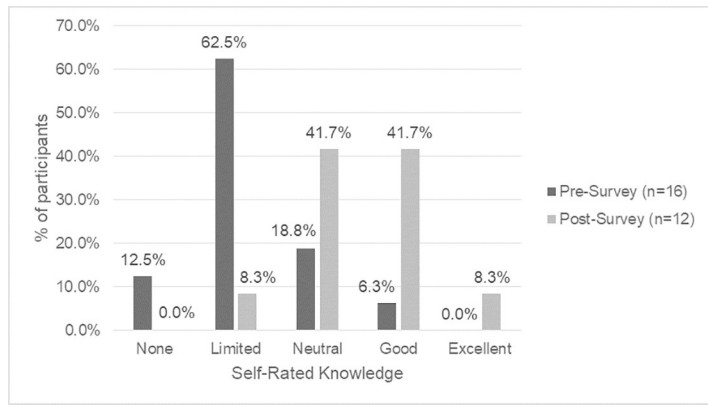

**Fig 2. Baseline vs. endline survey self-reported knowledge of Indigenous cultural protocols.**

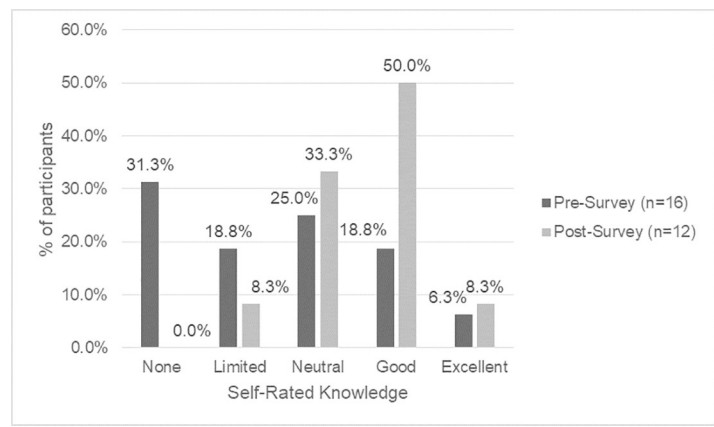

**Fig 3. Baseline vs. endline survey self-reported knowledge of OCAP® principles.**

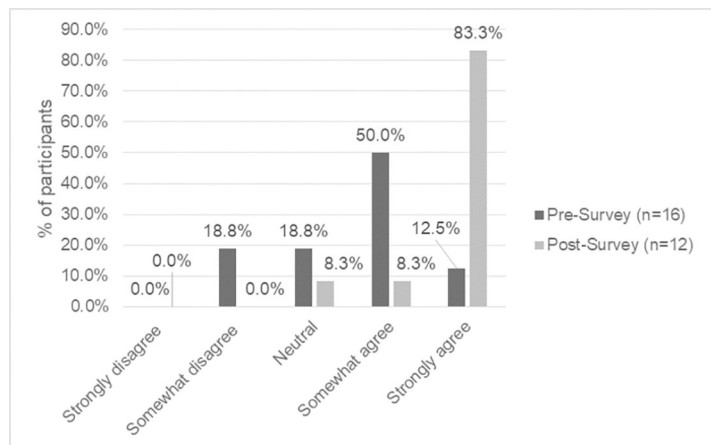

**Fig 4. Baseline vs. endline survey self-reported knowledge of Indigenous relationships to the land.**

Prior to the course, participants reported the highest level of knowledge in areas of colonization and its impacts on Indigenous issues (56.3% good, 0.0% excellent), with moderate understandings of Indigenous history in Canada (25.0% good, 0.0% excellent), terminology regarding Indigenous Peoples (25.0% good, 0.0% excellent) and OCAP® principles (18.8% good, 6.3% excellent). By the endline survey, more participants reported good knowledge in areas of colonization and its impacts on Indigenous issues (75.0% good, 16.7% excellent), Indigenous history in Canada (66.7% good, 0.0% excellent), and terminology regarding Indigenous Peoples (66.7% good, 8.3% excellent), with some reporting excellent knowledge. There was also an increase in understanding of Indigenous data sovereignty (baseline survey 0.0% good and 6.3% excellent vs. endline survey 58.3% good and 16.7% excellent).

In the endline survey only, the majority of participants indicated that their knowledge of Indigenous Traditional Knowledge after the program was good (58.3%) while others responded neutral (33.3%) or limited (8.3%). While most participant responses indicated that they had a good (75.0%) knowledge of the medicine wheel, there were also 8.3% of participants selecting excellent, and 16.7% of participants indicated limited knowledge.

Participants showed strong improvement in their knowledge of Indigenous relationships to the land and nature from baseline to endline, with 18.8% vs. 0% somewhat disagree, 18.8% vs. 8.3% neutral, 50.0% vs. 8.3% somewhat agree, and 12.5% vs. 83.3% strongly agree.

Participants also showed an increase in self-reported knowledge about local Indigenous protocols or practices and of local Indigenous communities from baseline (18.8% strongly disagree, 37.5% somewhat disagree, neutral 25.0%, 18.8% strongly agree) to endline (50.0% somewhat agree, 50.0% strongly agree).

## Reflection on privilege, positionality, and action

Participants were asked to reflect their social positionality in terms of race, class, gender, sexuality, and ethnicity and how it affects their work. The baseline survey asked if they reflected on social positionality and responses ranged from neutral to strongly agree (25.00% neutral, 43.8% somewhat agree, 31.3% strongly agree). In the endline survey, responses on if they intend to reflect on their social positionality shifted to somewhat agree (33.3%) but mainly strongly agree (66.7%).

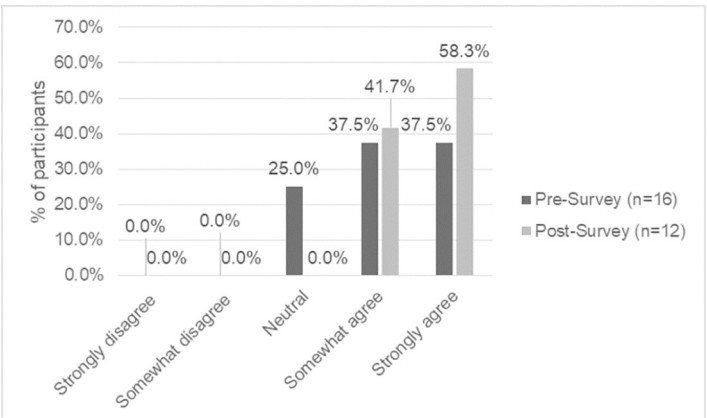

**Fig 5. Baseline vs. endline survey indication of reflection or intention to reflect on privilege and its impact on work.**

When asked to reflect on their privileges, including within their work (Fig 5), the responses in the baseline survey on if they do reflect on their privileges ranged from neutral to strongly agree (25% neutral, 37.5% somewhat agree, 37.5% strongly agree). The endline survey asks if they intend to reflect on their privilege and the responses were divided between somewhat agree (41.7%) and strongly agree (58.3%).

Building off social positionality and privileges, participants were asked in the baseline survey if they reflect on how to utilize their privilege to create social change for Indigenous Peoples and in the endline survey if they intend to regularly reflect on using privilege for social change. The baseline survey shows an array of responses from somewhat disagree to strongly agree (18.8% somewhat disagree, 37.5% neutral, 18.8% somewhat agree, 25.0% strongly agree). The endline surveys shows that all participants either somewhat agreed (50%) or strongly agreed (50%) that they intend to reflect regularly on how they can use their privilege to create social change.

Following the theme of social change for Indigenous Peoples, in the endline survey participants were asked if they feel more confident in supporting Indigenous Peoples in working towards equity and justice. The responses varied with 9.1% indicating that they somewhat disagreed, 45.5% indicating they somewhat agreed and 45.5% indicating that they strongly agreed.

Participants either somewhat agreed (66.7%) or strongly agreed (33.3%) when asked if they feel more confident speaking on Indigenous issues if they were asked to in the endline survey. Additionally, most participants indicated that they agreed that they were more aware of their positionality, powers, and privileges after the land-based learning experience (50.0% somewhat agree and 41.7% strongly agree). One participant (8.3%) somewhat disagreed that they were more aware of their positionality, powers, and privileges after the experience.

When asked to describe the impact of their learning on their relationships with people and surroundings, 2 people responded that there was no impact on their day to day. One participant said they were always respectful. Another participant said that they would use the knowledge in "certain contexts". The remaining participants were clear that there was much to continue to learn and use. The endline survey expands on these themes to include how individual positionality and privilege can be used to create social change and equity for Indigenous Peoples.

### Relationship with the physical and social environment

The baseline survey posed statements to the participant on their experience in nature. Most participants (strongly agree, 75%) indicated that they felt good about themselves after engaging in outdoor activities with 25% indicating that they somewhat agreed. Participants indicated that they mainly completed outdoor activities with other people (31.3% strongly agree and 50.0% somewhat agree). While most participants indicated that they enjoyed learning from others in a community setting (56.3% somewhat agree, 37.5% strongly agree), 6.3% of participants indicated that they somewhat disagreed with the statement.

In the baseline survey, most participants either strongly agreed (50.0) or somewhat agreed (37.5) that they had a clearer mind and made better decisions after engaging in outdoor activities. The endline survey asked if the land-based learning program gave them a sense of clarity with participants agreeing (50.0% somewhat agree and 41.7% strongly agree) and the other responses were neutral (8.3%).

Similarly, most participants selected that they strongly agreed (68.8%) that they felt relaxed after outdoor activities with 31.3% selecting somewhat agreed in the baseline survey. Participant response in the endline survey on if the experience made them feel more relaxed varied greatly with 8.3% strongly disagreeing, 8.3% somewhat disagreeing, 25.0% neutral, 25.0% somewhat agreeing, and 33.3% strongly agreeing.

Participants described their local environment in relation to beings in nature such as cultivating a green thumb or observing nature. Urban environments and the ecosystem juxtaposition of urban environments was described, one participant enjoyed the blend of urban and nature, while others felt less at home in urban environments and used nature as an escape from an urban environment. Water was an element that signified home and led to positive feelings of relaxation and nourishment. Community was mentioned in terms of the people (family friendly), the physical community (gratefulness to live in a neighborhood with greenery), diversity (cultural), and compromise (struggles living downtown but is safe). The theme of appreciation was also mentioned broadly, such as appreciation of nature, of feeling safe, of participating in outdoor activities.

The local environments that participants mentioned brings them joy include the cyclical nature of the environment such as the various stages of the life cycle in a dense forest, the rhythm and movement of water, and caring for plants. Locations associated with families were also environments that brought them joy and the reasoning was because of the tie to their families.

Participants expressed that natural environments offers peace it gives them a place to reflect, and that the sensory aspect of nature can also provide this peace. Being near water also brough joy as it is soothing. Lastly, the flora and fauna (beings) were also included as local environments that bring joy.

Participants were also asked to indicate their level of knowledge with regards to plants (25.0% limited, 33.3% neutral, 33.3% good, 8.3% excellent), trees (16.7% limited, 25.0% neutral, 41.7% good, 16.7% excellent), animals (25.0% limited, 41.7% neutral, 25.0% good, 8.3% excellent), the land (9.3% limited, 25.0% neutral, 50.0% good, 16.7% excellent), and water (16.7% limited, 16.7% neutral, 50.0% good, 16.7% excellent). In the endline survey, most participants agreed (66.7% strongly agree and 25.0% somewhat agree) that they have a greater appreciation for the Land and its gifts, with 8.3% responding neutral and (33.3% strongly agree and 58.3% somewhat agree) that they intent to apply the teachings received from Elders, Knowledge Keepers, and facilitators in their daily life, with 8.3% responding neutral.

Participants were asked how the experiences impacted their relationship with how they interacted with people and their surroundings. Different types of relationships that were

impacted such as with themselves (spending time with yourself), appreciation for others and appreciation for the land. Participants expressed how their idea of introductions changed and the shift towards a humanistic approach to feel connected to others and challenging professionalism. One participant wrote, "this experience gave me a sense of empathy, compassion, and the ability to ask questions but also more importantly listen". The vulnerability that comes with sharing personal stories and being authentic was acknowledged. Furthermore, participants expressed that they want to represent themselves and their culture authentically rather than fit in the Western framework.

The participants positive feelings associated with nature continued from the baseline to the endline survey. Additionally, the land-based learning program allowed participants to learn about different aspects of the environment and the land such as trees and water. This is demonstrated when the majority of participants agreed that they have a greater appreciation for the Land after the program.

## Reflections on the land-based learning experience

Prior to participating in the program, the majority of participants had not completed any similar Indigenous cultural safety trainings. Those who had completed formal cultural safety courses/training named the San'yas and OCAP trainings and some participants had attended short, informal workshops/trainings. Only one participant had been to a land-based programming before.

Factors that influenced participants' decision to register for the land-based learning event include personal (gain knowledge) and professional (applying cultural safety to their work and the intersection of Indigenous Peoples, cultures, and traditions with research). Participants wrote that the land-based learning experience provided the opportunity for skill improvements such as ensuring culturally safe interactions and research. It also was a space for them to broaden their knowledge, whether it be to strengthen their current knowledge, curiosity, or to build on previous cultural safety training. Lastly, the opportunity to build relationships was a factor in enrollment. The opportunity to learn from the Elders and Principal Investigator in-person, to appreciate the Earth, to foster community with other TRANSFORM HF members, to apply what was learnt at this event to support Indigenous Peoples in their own work, and to engage with Indigenous health advocates.

Participants hoped to gain connection, to reflect, to build on their research skills, and strengthen their knowledge and understanding. They hoped to create connections and build collaborations. In a professional capacity, there was interest in how to approach research collaborations and interact in a culturally safe way with Indigenous Peoples and to learn from Elders' experiences and how it pertains to proposed research. Being able to connect with other colleagues who are also interested in Indigenous cultural safety. In terms of building on their research skills, participants hoped to learn about Indigenous research methodologies, integrating strength-based approaches to research and learning processes, and how to improve research design. Lastly, participants hoped to strengthen their knowledge and understanding on Indigenous health, culture, and practices. As well as find commonalities within cultures and to increase their confidence surrounding Indigenous health.

Participants anticipated that the environment at Hart House Farm would provide a calming and reflective environment, would provide the opportunity for growth, would allow participants to be outdoors and with nature, and that it would facilitate connection with Elders and other participants. After completing the land-based learning experience, participants expectations ranged widely from being exceeded to not met. Participant expectations were exceeded due to the interactive elements of the program, the sharing of Traditional Knowledge, being

with the Elders, and the connection to the land. Reasoning for their expectations being met were that they experienced a positive learning environment, experienced community building and now have a new understanding and appreciation of Indigenous health and ways of knowing. Some were mixed as they "expected to be surrounded by Indigenous People and environment" and expected a different approach to learning (one that is more rigid or concrete). The experience did not meet expectations for those who expected less discussion and more time on the land, finding the talking circles to be a source of stress.

In the endline survey, participants were asked if something happened that they did not expect. Participants shared that the commonalities they saw between what was being discussed and their own values, histories and culture was not expected. Some participants were not sure what to expect from the program and so aspects of the program itself were unexpected including participating in ceremonies, the information itself, a passion for Indigenous health and the relationship between being on the land and connecting with knowledge being learnt. The sense of community within the TRANSFORM HF team that was fostered over the program was mentioned. It was also noted that the program was emotionally challenging and taxing. Similarly, the experience caused unexpected stress as the program was different than what was expected which was more focus on Elders and living on the land.

Participants felt more connected with TRANSFORM HF team after the land-based learning education program. In the baseline survey, the results varied with 6.3% responding somewhat disagree, 43.8% responding neutral, 31.3% responding somewhat agree, and 18.8% responding strongly agree. In the endline survey, participants mostly responded either somewhat agree (25.0%) or strongly agree (66.7%) with 8.3% responding neutral.

Finally, all participants wrote that they would participate again and some suggested additional activities for future events. As this was an introductory event, there was a baseline amount of content that had to be discussed. In future, more activities could be planned if participants were in residence or had a basic understanding of key issues related to research with Indigenous Peoples.

## Discussion

Being on-the-land and the physical environment at Hart House Farm is important for the program and the participants. In the baseline survey, the participants expressed their own positive relationships with their physical environment and the significance it held to them which provided a common ground for the land-based Indigenous cultural safety program. This program gave participants a unique perspective and experience to learn about Indigenous cultural safety which strengthened their own relationship with their physical environment but also provided a tangible connection to the material through the land and learning from Indigenous facilitators.

A culturally safe evaluation of a land-based program (Project Jewel) for Indigenous Peoples found that being on-the-land was a drive for participation [31]. Similarly, the results of our surveys reinforce the importance of land-based education. Our baseline survey shows that being on-the-land was a motivator for registration and in the endline survey, all participants indicated that they would attend another land-based experience. Reflections from first-year Bachelor of Science in Nursing students who participated in an Indigenous-led on-the-land cultural immersion that used Indigenous teaching methodology showed that this experience led to more awareness of "local Indigenous land and people" [32]. This is reflected in the endline survey as participants expressed an increased awareness on Indigenous history, culture, protocols, and practices. A cultural safety workshop for healthcare professionals that included Indigenous facilitators (i.e., Elder) found that having Indigenous facilitators was a strength

and the participants appreciated both the opening ceremony and the experiences the Indigenous facilitators shared [33]. Similarly, the participants from our land-based program shared that learning from Elders was a motivator for enrolling in the program but in the endline survey that hearing and learning from Indigenous facilitators was meaningful and appreciated.

After the land-based experience, participants indicated more robust knowledge of Indigenous histories, cultures, and practices as well as ethical conduct (e.g., OCAP®, data sovereignty, terminology). There was comparatively less knowledge of traditional medicines and cultural protocols among participants, but this was reflective of the lower knowledge at baseline on these topics as well as the discretion associated with sharing knowledge of traditional medicines and cultural protocols. Certain types of Indigenous knowledge are only appropriate to share in select settings (e.g., some stories are exclusively shared during wintertime). Overall, the increase in knowledge across a range of topics reflects a well-rounded foundation of information from which participants can continue their cultural safety journeys, in alignment with the advised content of training according to the Call to Action #23 and #24 of the Truth and Reconciliation Commission [TRC 15]. Further research could examine the differential effects of land-based learning for those with compared to those without previous cultural safety training.

By identifying and reflecting on their own power, positionality and privilege throughout the course, participants learned how to apply the knowledge they learned to create or further social change in the interest of equity and justice for Indigenous Peoples. The endline results demonstrate a shift from varied levels of reflection to a stronger level of intention for each of the themes (social positionality, privilege, and social change). Furthermore, most participants feel confident in supporting Indigenous Peoples towards equity and justice. The on-the-land programming education/lessons, both from the facilitator and Elders, drove participants to reflect internally, make connections with what they know/observe externally with relation to Indigenous Peoples and the importance of creating social change.

An evaluation by Mills and Creedy [34] on emotion-based pedagogical intervention for cultural safety, the authors found that even though the undergraduate students had emotional learning experiences, short and long-term application of the principles and teachings were not indicated with specificity "in a practical and meaningful way". Similarly, many TRANSFORM HF participants identified strong emotions (e.g., peaceful, emotionally touching, inspired, grateful, stressed) throughout their land-based learning experience, but the articulation of short and long-term application of their learning was limited. The concept of "human introductions" was a commonly referenced takeaway from to the program but there was no mention of how they would apply culturally safe practices at work or their intentions on decolonization and anti-racism. Additionally, four participants did not feel or see that the experience impacted their relationships. Yaphe et al. [33] noted that the difficult content of cultural safety training is "a tough pill to swallow" and that there was apprehension which led to some participants to be dissatisfied. Feeling uncomfortable and difficulty with the material is a part of growth in learning about cultural safety [33]. This is similarly reflected in our findings as participant responses were varied when indicating if the on-the-land program made them feel relaxed and that an unexpected aspect was that the program did cause stress and was emotionally taxing.

Feedback from participants on the land-based experience reflected a need for further research on the effect of land-based learning as cultural safety training across different learning styles. While there are a limited number of evaluations and publications on land-based learning as cultural safety training [26, 27, 31], TRANSFORM HF participants expressed a range of opinions on the format of land-based learning, including the use of talking circles. The rigidity and predictability of classroom or lecture-style learning may be more comfortable for those

who have not had interactive classes, but comparative efficacy is unknown. Being on the land and engaged in talking circles is important for building community and putting knowledge into practice but can be stressful for participants who are more introverted, have difficulty with publicly speaking, or do not have similar (Indigenous) cultural values that support sharing in a community environment. As there were only Indigenous instructors (no participant self-identified as Indigenous), the format of learning would be jarring compared to webinars, lectures, or book-learning. It has been noted that dissatisfaction in cultural safety training could be due to the difficulty of the material or the training being perceived as a critique, however, the discomfort is important for change and participants could be made aware of the nature of the content beforehand [33].

Not surprising, all the participants wanted another on-the-land event. As this was an introductory event, there was a lot of history to attend to. In future, more activities could be planned if participants were in residence or had a basic understanding of key issues related to research with Indigenous Peoples. While many participants requested more activities, this requires that it doesn't become voyeuristic or tourist-y. To prevent this, there must be reciprocal work done which is hard to do at the Hart House Farm as it is not Indigenous-led. Reciprocity requires that after learning from the land and Elders, that you return the "favour" by providing some resource/assistance/support to Indigenous Peoples. If this were in a First Nation or Indigenous organizational space, then assistance could be rendered, but the University of Toronto is a colonial university property so any reciprocity goes to the colonial institution.

## Limitations

The length of the program (3 days) and lack of sustained immersion (day-program format) limited the amount, range and depth of content that could be presented, and the sharing of experiences both by participants and speakers. However, given the increase in self-reported knowledge; reflection; intention to change; and relationships with people and land reported by participants, the short period of time was sufficient to create self-perceived movement towards cultural safety.

Another limitation is that the evaluation was short-term and based on participants' self-report. Since the endline survey opened as soon as the program was finished, there was no long-term evaluation of the impact of the training. Additionally, the self-report format is influenced by participants' self-perception, and may be subject to social desirability bias, leading them to overestimate their growth to appear as 'good students' [35–37].

The small group size, while an intentional design choice, also limited the study findings. The number of participants who completed the endline survey (n = 12) was lower than the number of participants who completed the baseline survey (n = 16). Furthermore, only 11 participants completed question 4F. However, the sample size was intentionally small to encourage the creation of a cohort or community.

## Conclusion

Cardiovascular disease disproportionately impacts First Nations, Inuit and Métis communities in Canada, and there is a need for cultural safety training among professionals (staff and trainees) who are creating digital health tools, programs or providing care to ensure the availability of safe, effective, and respectful cardiac care. Land-based approaches center Indigenous ways of knowing and being through the learning process but have not been widely evaluated as a cultural safety training among non-Indigenous people. In this evaluation, participants in the land-based experience reported improved knowledge, awareness, and self-reflection, which are each important components of cultural safety, as well as a strengthened relationship with

their physical environment. A key takeaway among participants was having a human-focused outlook on their professional relationships moving forward. Overall variation in responses indicated that participants are still beginning their (un)learning journey and that further reflection is necessary for long-term and specific implementations of the teachings and knowledge. There is a need for future research to add to this preliminary research on the impact of land-based learning as a form of cultural safety training, and examine the long-term impacts on health care interactions and outcomes for Indigenous Peoples.

## Acknowledgments

We thank Elders Ellisa Johnson, Luana Harper, and Scott Debiassage for sharing their experiences and guiding the participants through the program content. We also thank Augusta Lipscombe from TRANSFORM HF for their work in coordinating and planning the event.

## Author Contributions

**Conceptualization:** Angela Mashford-Pringle.

**Data curation:** Javiera-Violeta Durán Kairies, Emma J. Rice.

**Formal analysis:** Javiera-Violeta Durán Kairies, Emma J. Rice.

**Funding acquisition:** Sterling Stutz, Sharon W. Y. Tan, Anne Simard, Heather Ross, Angela Mashford-Pringle.

**Investigation:** Sterling Stutz, Angela Mashford-Pringle.

**Methodology:** Anne Simard, Heather Ross, Angela Mashford-Pringle.

**Project administration:** Sterling Stutz, Sharon W. Y. Tan.

**Resources:** Angela Mashford-Pringle.

**Software:** Javiera-Violeta Durán Kairies, Emma J. Rice.

**Supervision:** Anne Simard, Heather Ross, Angela Mashford-Pringle.

**Validation:** Angela Mashford-Pringle.

**Visualization:** Javiera-Violeta Durán Kairies, Emma J. Rice.

**Writing – original draft:** Javiera-Violeta Durán Kairies, Emma J. Rice.

**Writing – review & editing:** Javiera-Violeta Durán Kairies, Emma J. Rice, Sterling Stutz, Anne Simard, Heather Ross, Angela Mashford-Pringle.

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
