## [Decision Letter · Decision Letter 0]

12 Mar 2024

PONE-D-23-30936Transform[ing] Heart Failure Professionals with Indigenous Land-Based Cultural Safety in Ontario, CanadaPLOS ONE

Dear Dr. Mashford-Pringle,

Thank you for submitting your manuscript to PLOS ONE. After careful consideration, we feel that it has merit but does not fully meet PLOS ONE’s publication criteria as it currently stands. Therefore, we invite you to submit a revised version of the manuscript that addresses the points raised during the review process.

We look forward to receiving your revised manuscript.

Kind regards,

Julio Cesar Ossa, Ph.D

Academic Editor

PLOS ONE

 [The On-the-Land Indigenous Education Program was funded by 2021 TRANSFORM HF Collaboration Starter Grant awarded to the P.I., Dr. Mashford-Pringle. TRANSFORM HF is funded by the Ted Rogers Centre for Heart Research and the University of Toronto [see https://transformhf.ca/]].  

5. We note that you have indicated that there are restrictions to data sharing for this study. PLOS only allows data to be available upon request if there are legal or ethical restrictions on sharing data publicly. For more information on unacceptable data access restrictions, please see http://journals.plos.org/plosone/s/data-availability#loc-unacceptable-data-access-restrictions. 

Additional Editor Comments:

It is recommended that the authors take into account the six observations made by the reviewers. This is an adjustment to the article that can improve its quality and make it more coherent according to their perspective of working with an indigenous population.

The manuscript presents a very interesting case for Indigenous land-based learning. I think the topic is worth being published but I suggest some major improvements before acceptance. This paper has a relevant theme, namely, the relationship between health and cultural practices. It analyzes the participants' impressions of the importance of the learning program described in the text. The approach to the results favors quantitative data, although there is no statistical treatment, since the sample size does not allow such an analysis.

1. I suggest an analysis of the discourse of health professionals on issues related to the specific health of indigenous people before and after the training. I think it would be a rich form of analysis and would complete the article.

2. In this way they would have elements to elaborate conclusions in a more conducive and more articulated way with a cultural epistemology and not from an external point of view.

3. A few more words on the Indigenous land-based approach, for the reader who is not familiar

4. Bit more details about participants background, about the learning activities

5. The pre post surverys descriptive statistics are not particularly interesting or informative. I understand that in the attempt to evaluate the method, it is necessary to do some surveys. However the most important part would be how participants reported their experience qualitatively. My suggestion is to sum up the descriptives in just 1 or 2 graphs max (by the way all the necessary information must be provided in the graphs, such as N=). Then discuss for instance how the medical or enginneeer epistemological framework negotiated with the indigenous one. Medical approach is a postivist, causalitic understandning. How did they really interact with the indigenous epistemology/cosmology? Or are you just looking for a behavioral effect? Just doing things differently when dealing with first nations?

6. Conclusions are a bit weak and leave a sense of "so what"? can be elaborated

Reviewers' comments:

Reviewer's Responses to Questions

**Comments to the Author**

1. Is the manuscript technically sound, and do the data support the conclusions?

Reviewer #1: Partly

Reviewer #2: Partly

2. Has the statistical analysis been performed appropriately and rigorously? 

Reviewer #1: N/A

Reviewer #2: N/A

3. Have the authors made all data underlying the findings in their manuscript fully available?

Reviewer #1: No

Reviewer #2: Yes

4. Is the manuscript presented in an intelligible fashion and written in standard English?

Reviewer #1: Yes

Reviewer #2: Yes

5. Review Comments to the Author

Reviewer #1: The manuscript presents a very interesting case for Indigenous land-based learning. I think the topic is worth being published but I suggest some major improvements before acceptance.

1) A few more words on the Indigenous land-based approach, for the reader who is not familiar

2) Bit more details about participants background, about the learning activities

3) The pre post surverys descriptive statistics are not particularly interesting or informative. I understand that in the attempt to evaluate the method, it is necessary to do some surveys. However the most important part would be how participants reported their experience qualitatively. My suggestion is to sum up the descriptives in just 1 or 2 graphs max (by the way all the necessary information must be provided in the graphs, such as N=). Then discuss for instance how the medical or enginneeer epistemological framework negotiated with the indigenous one. Medical approach is a postivist, causalitic understandning. How did they really interact with the indigenous epistemology/cosmology? Or are you just looking for a behavioral effect? Just doing things differently when dealing with first nations?

4) what is the understanding of hearth deseases in indigenous medicine?

5) conclusions are a bit weak and leave a sense of "so what"? can be elaborated

Reviewer #2: This work has a relevant subject, i.e., the relations between health and cultural practices. It analyzes the impressions of the participants about the importance of the learning program that was described in the text. The approach of the results privileges the quantitative data, even that it hasn't statistical treatment, because the size of the sample doesn't for an analyzis such that. Personally - and it says about my formation as a qualitative researcher - I would be more interested in the discourse of health professionals on issues involving the specific health of indigenous people before and after the training. I think that this would be a rich way of analyzis (that the authors may explore in other studies). As far as this study is concerned, I can find no reason to reject it. On the contrary, the relevance of the topic is adequately justified and the conclusions are based on the results described.

6. PLOS authors have the option to publish the peer review history of their article (what does this mean?). If published, this will include your full peer review and any attached files.

Reviewer #1: No

Reviewer #2: **Yes: **Filipe Degani-Carneiro

---

## [Author Response · Author response to Decision Letter 0]

3 Apr 2024

We have provided a response to all Editor and Reviewer comments.

---

## [Editor Report · Decision Letter 1]

15 Apr 2024

Transform[ing] Heart Failure Professionals with Indigenous Land-Based Cultural Safety in Ontario, Canada

PONE-D-23-30936R1

Dear Dr. Mashford-Pringle,

We’re pleased to inform you that your manuscript has been judged scientifically suitable for publication and will be formally accepted for publication once it meets all outstanding technical requirements.

Kind regards,

Julio Cesar Ossa, Ph.D

Academic Editor

PLOS ONE

Additional Editor Comments (optional):

Although there is no in-depth development of the reviewers' indications, I understand that this was not the original purpose of the article. Therefore, I consider that the reviewers' requirements have been met, since the adjustments that were requested were minor in nature, and this was the result of the reviewer's request.
---

## [Editor Report · Acceptance letter]

13 May 2024

PONE-D-23-30936R1 

PLOS ONE

Dear Dr. Mashford-Pringle, 

I'm pleased to inform you that your manuscript has been deemed suitable for publication in PLOS ONE. Congratulations! Your manuscript is now being handed over to our production team.

Kind regards, 

on behalf of

Dr. Julio Cesar Ossa 

Academic Editor

PLOS ONE